# Enteric nervous system modulation of luminal pH modifies the microbial environment to promote intestinal health

**M. Kristina Hamilton**[1,2], **Elena S. Wall**[2], **Catherine D. Robinson**[2], **Karen Guillemin**[2,3]*, **Judith S. Eisen**[1]*

**1** Institute of Neuroscience, University of Oregon, Eugene, Oregon, United States of America, **2** Institute of Molecular Biology, University of Oregon, Eugene, Oregon, United States of America, **3** Humans and the Microbiome Program, CIFAR, Toronto, Ontario, Canada

* kguillem@uoregon.edu (KG); eisen@uoregon.edu (JSE)

**Data Availability Statement:** All relevant data are within the paper and its Supporting Information.

**Funding:** Research reported in this publication was supported by the National Institutes of Health

## Abstract

The enteric nervous system (ENS) controls many aspects of intestinal homeostasis, including parameters that shape the habitat of microbial residents. Previously we showed that zebrafish lacking an ENS, due to deficiency of the *sox10* gene, develop intestinal inflammation and bacterial dysbiosis, with an expansion of proinflammatory *Vibrio* strains. To understand the primary defects resulting in dysbiosis in *sox10* mutants, we investigated how the ENS shapes the intestinal environment in the absence of microbiota and associated inflammatory responses. We found that intestinal transit, intestinal permeability, and luminal pH regulation are all aberrant in *sox10* mutants, independent of microbially induced inflammation. Treatment with the proton pump inhibitor, omeprazole, corrected the more acidic luminal pH of *sox10* mutants to wild type levels. Omeprazole treatment also prevented overabundance of *Vibrio* and ameliorated inflammation in *sox10* mutant intestines. Treatment with the carbonic anhydrase inhibitor, acetazolamide, caused wild type luminal pH to become more acidic, and increased both *Vibrio* abundance and intestinal inflammation. We conclude that a primary function of the ENS is to regulate luminal pH, which plays a critical role in shaping the resident microbial community and regulating intestinal inflammation.

## Author summary

The intestinal microbiota is an important determinant of health and disease and is shaped by the environment of the intestinal lumen. The nervous system of the intestine, the enteric nervous system (ENS), helps maintain many aspects of intestinal health including a healthy microbiota. We used zebrafish with a genetic mutation that impedes ENS formation to investigate how the ENS prevents pathogenic shifts in the microbiota. We found that mutants lacking an ENS have a lower luminal pH, higher load of pathogenic bacteria, and intestinal inflammation. We showed that correcting the low pH, using the commonly prescribed pharmacological agent omeprazole, restored the microbiota and prevented intestinal inflammation. Conversely, we found that lowering the luminal pH of wild type

(https://www.nih.gov/) under award numbers P50GM098911, P01GM125576, and P01HD22486 to KG and JSE, American Cancer Society (https://www.cancer.org/) Postdoctoral Fellowship to MKH (grant number PF-17-208-01-MPC), and a John Simon Guggenheim Memorial Foundation Fellowship (https://www.gf.org/) to JSE. The funders had no role in study design, data collection and analysis, decision to publish, or preparation of the manuscript. ACS and NIH P01HD22486 paid portions of MKH's salary. NIH P01HD22486 and Guggenheim paid portions of JSE's salary. NIH P50GM098911 and NIH P01GM125576 paid portions of KG's salary. NIH P01GM125576 paid EW's salary and portions of CDR's salary.

**Competing interests:** The authors have declared that no competing interests exist.

animals, using the drug acetazolamide, caused expansion of pathogenic bacteria and increased intestinal inflammation. From these experiments, we conclude that a primary function of the ENS is to maintain normal luminal pH, thereby constraining intestinal microbiota community composition and promoting intestinal health.

## Introduction

The enteric nervous system (ENS) was first identified in 1755 by Von Haller [1] for its role in intestinal motility but is now known to be responsible for many aspects of intestinal physiology [2]. This complex network of intestinal neurons and glia regulates nutrient absorption [3], barrier function [4], and waste clearance [5,6]. Zebrafish with a homozygous null mutation in the *sox10* gene, encoding a SRY-related HMG-box family transcription factor, lack an ENS due to the migration failure of neural crest cells, which give rise to enteric neurons and pigment cells [7,8]. Using the *sox10* mutant, we discovered that the ENS constrains intestinal microbiota composition and buffers against intestinal inflammation, as measured by intestinal neutrophil infiltration [9]. This inflammation was alleviated in the absence of the microbiota, showing that it is microbially induced, or by transplantation of wild-type (WT) enteric neurons into mutant hosts, showing that it is regulated by the ENS. To understand which host-associated microbes contribute to the inflammatory process, we profiled the microbiota and found expansion of proinflammatory *Vibrio* strains in *sox10* deficient intestines. Both the hyperinflammation phenotype and the abundance of proinflammatory bacteria varied across *sox10* individuals, which had a broader but overlapping distribution with WT siblings, with some *sox10* animals displaying normal abundances of neutrophils and *Vibrio*, whereas other *sox10* mutants exceeded the WT range in both [9]. The specific mechanisms by which the ENS shapes the intestinal environment to constrain microbial community membership and abundance remain to be determined [10].

Human *SOX10* mutations result in Hirschsprung disease (HSCR), which is characterized by ENS reduction and intestinal dysmotility [11,12]. HSCR is a phenotypically complex disease linked to multiple genetic loci, with variable disease severity even among carriers of the same disease gene allele [13]. A common complication of HSCR is Hirschsprung-associated enterocolitis (HAEC). Patients with this life-threatening inflammatory complication have altered microbiota, referred to as dysbiosis, compared to HSCR patients without enterocolitis [14–17]. How the altered intestinal physiology of HSCR patients affects the microbiota remains unknown [15,18,19]. Here, we report on our investigation of how ENS modulation of intestinal physiology establishes a normal luminal environment for the microbiota and the consequences of ENS absence on that environment.

In addition to HAEC, dysbiotic, proinflammatory intestinal bacterial communities have been described in a wide array of gastrointestinal disorders including inflammatory bowel disease (IBD). In these disorders, the complex interactions between host genetics, microbes, diet, and other environmental factors make it difficult to disentangle whether shifts in intestinal microbiota composition are a cause or consequence of intestinal inflammation [20]. Chronic intestinal pathological inflammation, as seen in IBD [21] and other conditions of "unresolving inflammation" [22], are characterized by an influx of immune cells, cytokines, and damage in the affected tissue [23]. During homeostasis, the intestine acts as both a physical epithelial barrier and an immunological barrier that together separate the intestinal lumen from the inside of the animal host. These barriers prevent unwanted, potentially toxic components from entering the blood stream. During pathological inflammation, intestinal barrier function is often

breached, resulting in intestinal hyperpermeability. The increased flux of molecules crossing the barrier, commonly referred to as "leaky gut," results in immune cell recruitment to the breach [24,25]. Inflammation and intestinal hyperpermeability thus create a positive feedback loop [26–30], making it challenging to identify the primary driver of intestinal pathology. The ENS has been suggested to influence intestinal permeability [31,32] and hyperpermeability in HSCR patients has emerged as a predictive factor of patients' risk for postoperative complications and enterocolitis [33]. Experimental models in which host and microbial factors can be independently manipulated are needed to identify primary defects that lead to establishment of proinflammatory, dysbiotic, intestinal microbial communities.

The zebrafish model provides an ideal opportunity to identify host features that maintain intestinal microbiota homeostasis. The development, tissue organization, and physiology of the zebrafish intestine is similar to the mammalian gastrointestinal tract, although zebrafish, like many fishes, lacks a stomach with specialized acid-secreting cells [34,35]. Zebrafish intestinal microbiota are well-characterized, with individual members exhibiting a range of immunomodulatory activities, both pro- and anti-inflammatory [36,37]. We can easily derive hundreds of genetically related individuals germ-free (GF), a state in which they are devoid of intestinal microbes and lack intestinal inflammation, enabling us to identify primary host defects that drive inflammatory processes. Moreover, larvae are nearly transparent, facilitating live imaging [38,39]. Also, zebrafish are tractable to genetic manipulations, providing insights into normal ENS development, function, and human diseases that result when these processes go awry [40]. We have shown that ENS controlled intestinal physiology has significant effects on zebrafish microbiota community dynamics, specifically influencing competition between commensal community members, *Vibrio* and *Aeromonas* [41,42]. Importantly, *sox10* mutants lacking an ENS exhibit intestinal expansion of proinflammatory members of the *Vibrio* genus compared to their co-housed WT siblings, thus providing an opportunity to investigate how intestinal microbiota communities differ between hosts within the same environment.

Dysbiosis can be challenging to study because it is often characterized by accentuated inter-individual variability in microbiome composition among affected individuals [43]. The high fecundity and inexpensive husbandry of zebrafish facilitate well-powered studies with extensive replication to reveal relationships between variable microbiomes and host disease phenotypes. Here we use conventionally reared, germ-free, and mono-associated zebrafish to investigate characteristics of the *sox10* mutant intestine that make it permissive for *Vibrio* overgrowth. We show that intestinal transit, intestinal permeability, and luminal pH regulation are aberrant in *sox10* mutants. Contrary to our initial hypothesis that these changes result from microbial dysbiosis, we find that they occur in the absence of microbes, and thus are primary defects of ENS loss that are independent of microbially induced inflammation. Moreover, we find that ENS-mediated luminal pH regulation is both necessary and sufficient to modulate proinflammatory bacterial overgrowth. Thus, in *sox10* mutants, absence of the ENS causes luminal pH to become more acidic, resulting in proinflammatory bacterial overgrowth and intestinal inflammation.

## Results

### The ENS promotes intestinal transit and prevents intestinal hyperpermeability independent of microbially induced inflammation

*sox10^t3^* mutants completely lack an ENS, resulting in altered intestinal motility and microbially induced inflammation [9,44]. We validated these previous results by measuring intestinal neutrophil abundance and rate of intestinal transit at 6 days post fertilization (dpf). As expected, conventionally (CV) raised *sox10* mutants showed a significant increase in intestinal

neutrophils compared to CV WTs (Fig 1A–1C). To confirm that this hyperinflammation was microbially induced, we derived *sox10* mutants and their WT siblings GF and found that GF WT and *sox10* mutants both had fewer neutrophils than CV WTs (Fig 1C). Intestinal transit

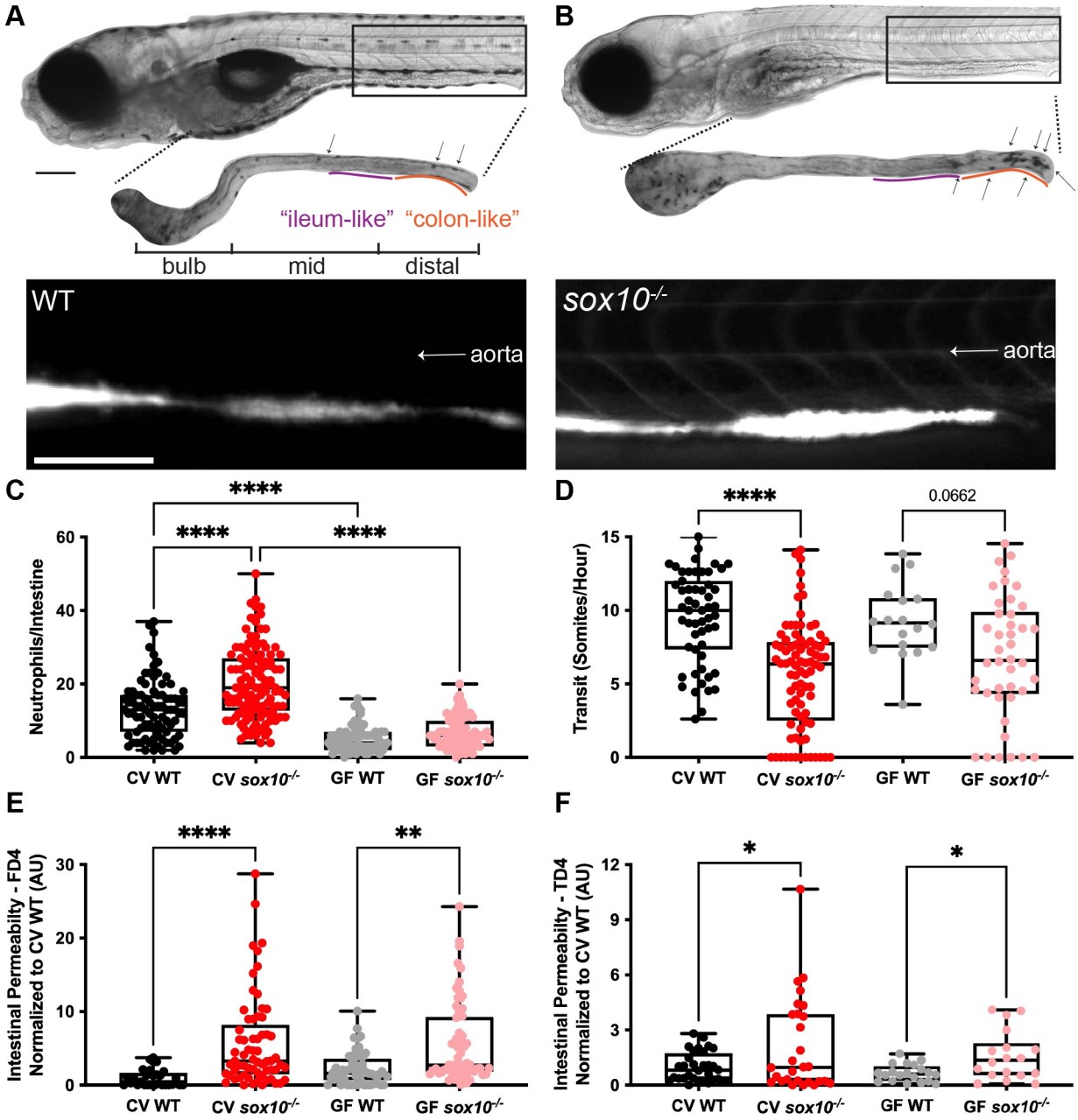

**Fig 1. *sox10* mutants have decreased intestinal transit and hyperpermeability, independent of intestinal microbially induced inflammation.** Representative bright field and fluorescence microscopy images showing (A) WT, and (B) *sox10*<sup>-/-</sup> larvae (top), dissected intestines stained for myeloperoxidase activity to reveal intestinal neutrophils (middle) and after oral microgavage of 4 kD fluorescein dextran (FD4) (bottom). Arrows indicate intestinal neutrophils. (C) Quantification of intestinal neutrophils per mid and distal intestine in CV and GF larvae. (D) Quantification of intestinal transit of phenol red; distance traveled over time measured as somites/hour (GF WT vs GF *sox10*<sup>-/-</sup> t-test p = 0.02). Quantification of (E) FD4 and (F) 4 kD tetramethylrhodamine dextran (TD4) permeability measured as fluorescence intensity in aorta. Scale bars = 300μm. AU = arbitrary units. Each dot in C-F is one fish; n>18 for each condition. Boxes in C-F represent the first to third quartiles, center bar denotes the median, and whiskers the maximum and minimum of each dataset. * p < 0.05, **p<0.01, ****p<00001. A and B: ANOVA followed by Tukey's post hoc test, C and D: Kolmogorov-Smirnov non-parametric t-test.

was determined by gavaging [45] larvae with phenol red and recording the location of the most distal extent of dye both immediately following gavage and 40–50 min later. These experiments revealed that CV *sox10* mutants had decreased intestinal transit compared to their CV WT siblings. In contrast to the neutrophil phenotype, this decreased transit in *sox10* mutants versus WT siblings still occurred in the GF state (Fig 1D; Student's t-test p = 0.02; see S1 Data File for all statistical tests). These data demonstrate that the ENS promotes intestinal transit independently of the microbiota whereas hyperinflammation depends on microbiota presence. We explored possible relationships between transit rate and inflammation in the different genotypes and microbial states (S1A and S1B Fig). We observed that in CV WTs, neutrophil number tended to be higher in individuals exhibiting slower transit. Neutrophil number and transit rate were also significantly correlated in GF WTs, even when overall neutrophil numbers were low (S1A Fig). High neutrophil numbers also trended toward slower transit times in CV WT animals (S1A Fig). In contrast, in *sox10* mutants this relationship between transit rate and neutrophil number was not maintained, suggesting other factors may explain variation in *sox10* mutant intestinal hyperinflammation (S1B Fig).

The ENS has been suggested to play a role in the intestinal barrier function that regulates intestinal epithelium permeability [4]. Because intestinal permeability is closely linked to intestinal inflammation and microbial dysbiosis, we hypothesized that the intestinal barrier would be hyperpermeable in the absence of the ENS [46]. To test this hypothesis, we quantified intestinal permeability by gavaging small fluorescent molecules, fluorescein isothiocyanate-conjugated (FITC)-dextran (FD4, 4 kD; Fig 1E) or tetramethylrhodamine isothiocyanate (TRITC)-dextran (TD4, 4 kD; Fig 1F) [30,45], into intestines of 6 dpf WTs and *sox10* mutants and measuring fluorescence in the vasculature *in vivo* (Fig 1A). CV WTs did not have detectable fluorescence in the vasculature by eye (Fig 1A). In contrast, CV *sox10* mutants had increased levels of fluorescence in the aorta compared to CV WTs (Fig 1E and 1F), visible by eye (Fig 1B). We noted considerable variation in aorta fluorescence among CV *sox10* mutants, some of which did not exhibit hyperpermeability. This level of variation in hyperpermeability is similar to the variation in inflammation we previously described in CV *sox10* mutants [9]. To determine whether this hyperpermeability could be the cause or consequence of microbially induced inflammation, we measured intestinal permeability in *sox10* mutants and their WT siblings that had been derived GF. We observed that even in the absence of microbiota, GF *sox10* mutants had significantly higher levels of fluorescence in the aorta compared to GF WTs via the Kolmogorov-Smirnov non-parametric test (FD4 p<0.0001; TD4 p = 0.025) (Fig 1E and 1F). These results show that intestinal hyperpermeability occurs in *sox10* mutants independently of the microbiota and microbially induced inflammation, suggesting that hyperpermeability is a primary defect that may prime the luminal environment for conditions leading to microbial dysbiosis and microbially induced inflammation in *sox10* mutants. Interestingly, permeability and neutrophil number were uncorrelated in both WTs and *sox10* mutants (S1C and S1D Fig), yet slower transit times were significantly positively associated with increased intestinal permeability in the same animals (S1E Fig). These associations suggest that in a normally functioning intestine with microbial colonization, slower transit is associated with increased intestinal neutrophil numbers and permeability. Together, these data show that decreased intestinal transit and increased intestinal permeability in *sox10* mutants occur independently of microbially induced inflammation.

To learn whether the decreased transit and intestinal hyperpermeability seen in *sox10* mutants are primary defects that lead to intestinal inflammation, we asked whether these defects precede intestinal inflammation by measuring intestinal neutrophil number, permeability, and transit at 4, 5, 6, and 7 dpf. We found that neither neutrophil number nor permeability was increased in *sox10* mutants at 4 dpf. Intestinal permeability was significantly greater

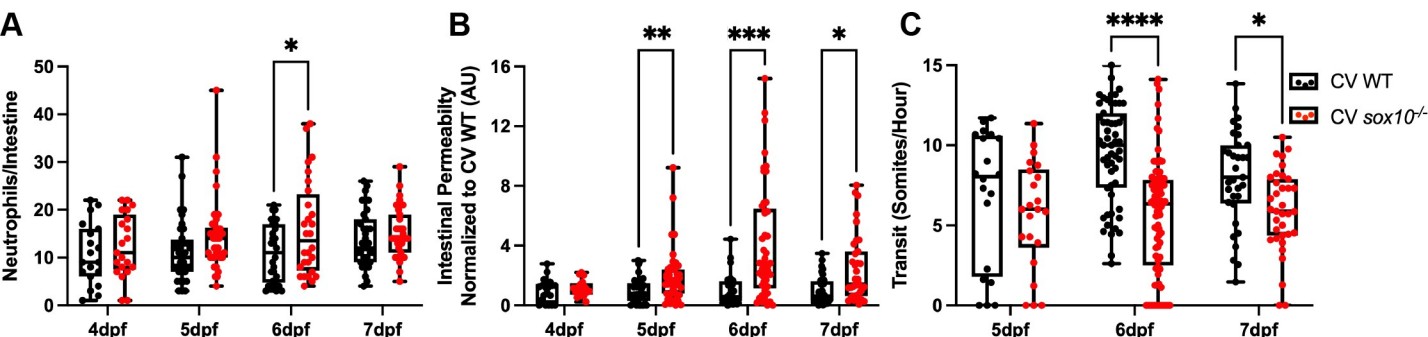

**Fig 2. Hyperpermeability in CV *sox10* mutants precedes decreased transit and microbial induced inflammation.** (A) Quantification of intestinal neutrophil response. (B) FD4 permeability as measured by fluorescence intensity in aorta. (C) Transit at 4, 5, 6, and 7 dpf. Each dot is a fish; n>18 for each condition. Boxes represent the first to third quartiles, center bar denotes the median, and whiskers the maximum and minimum of each dataset. $^*$ p < 0.05, $^{**}$p<0.01, $^{****}$p<00001. A&C: Two-way ANOVA followed by Šídák multiple comparisons test; B: Kolmogorov-Smirnov non-parametric t-test.

in *sox10* mutants compared to WTs at 5 dpf (Fig 2B), whereas neutrophil number and transit rate were not significantly different from WT siblings until 6 dpf (Fig 2A and 2C). Intestinal permeability remained increased, and transit remained decreased in *sox10* mutants compared to WT at 7 dpf (Fig 2B and 2C). The number of neutrophils in WT increased to levels comparable to co-housed *sox10* mutant siblings by 7 dpf (Fig 2A), which could be due to increased levels of transmission of proinflammatory microbes from dysbiotic *sox10* individuals to WTs. The observation that hyperpermeability in *sox10* mutants is independent of microbiota and preceded intestinal neutrophil influx by at least a day suggests that ENS-mediated regulation of permeability is required to prevent microbial dysbiosis and intestinal inflammation.

## The ENS promotes intestinal tight junction integrity

Hyperpermeability has been attributed to altered intestinal tight junctions, and measuring intestinal permeability via the fluorescent marker FD4 (see above) is widely used to assess the paracellular leak pathway specifically [47]. Tight junctions are the primary structural component regulating paracellular permeability [48] and the ENS has been suggested to regulate tight junction proteins [32], specifically ZO1 and Occludin [49]. Accordingly, we asked whether intestinal tight junctions are altered in *sox10* mutants relative to WTs. Regionalization of the larval zebrafish intestine is well conserved with the mammalian intestine [35]. In rodents, intestinal permeability is regionally dependent [50], therefore we analyzed ZO1 and Occludin protein localization in two regions (see Fig 1A), the "ileum-like" region (about 400 μm rostral of the anus, called the vent in fishes) and the "colon-like" region (about 100 μm rostral of the vent). Endocytosis that decreases Occludin binding to ZO1 is a signature of the paracellular leak pathway and macromolecular barrier loss [51], thus we measured colocalization of ZO1 and Occludin. Colocalization was similar between CV WTs and *sox10* mutants in the ileum-like region, however colocalization was significantly lower in GF *sox10* mutants compared to CV *sox10* mutants (Fig 3A and 3B). This result indicates that tight junction integrity in the ileum-like region is independent of the ENS and the microbiota, yet the microbiota can influence ZO1 and Occludin colocalization when the ENS is not present. Importantly, colocalization of ZO1 and Occludin was decreased in both CV and GF *sox10* mutants in the colon-like region compared to their WT siblings (Fig 3C and 3D). This is consistent with the intestinal hyperpermeability being due to absence of the ENS and independent of the presence of microbiota.

Claudin proteins are widely expressed at intestinal epithelial tight junctions and Claudin-2 specifically regulates the paracellular pore pathway responsible for ion secretion in mammals

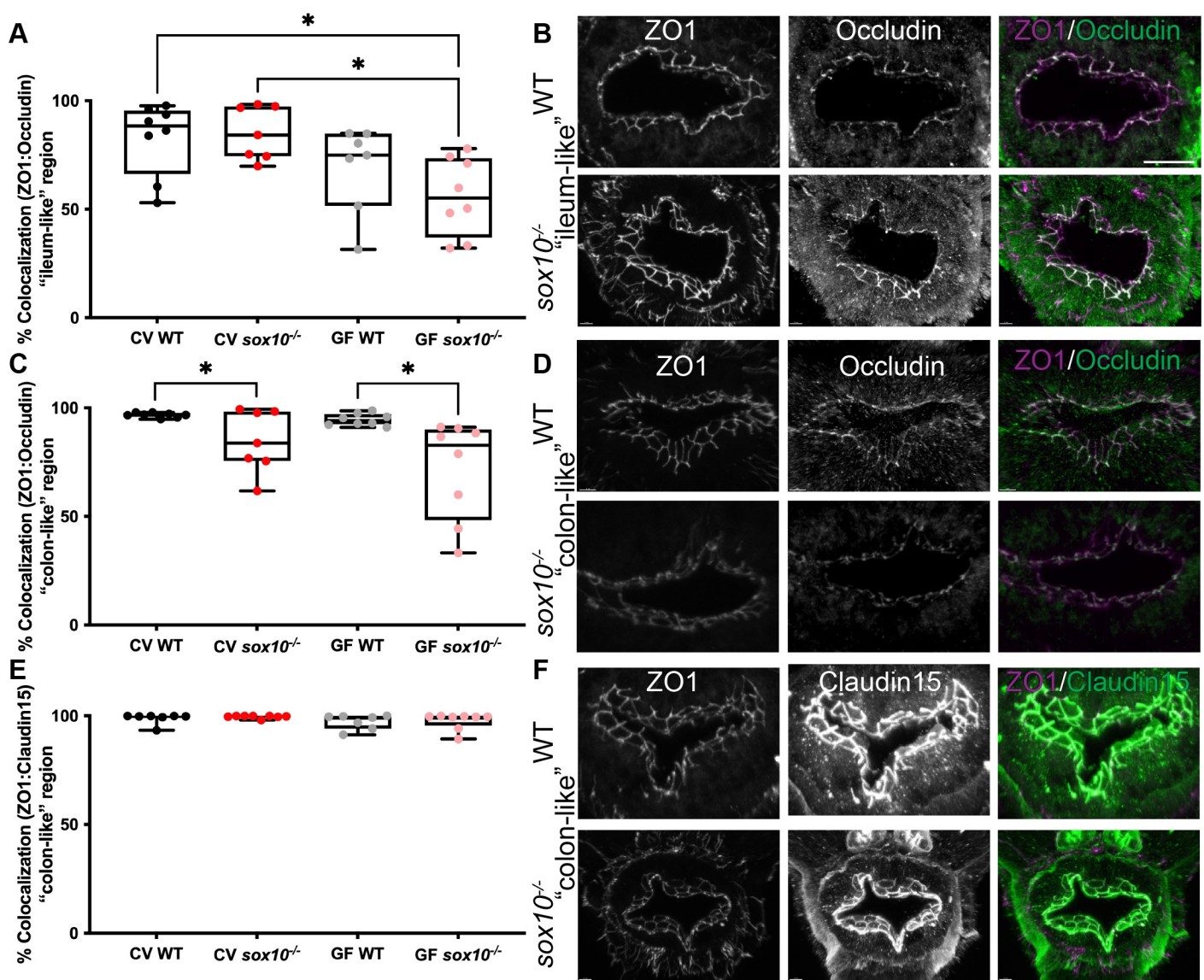

**Fig 3. *sox10* mutants have altered intestinal tight junctions.** (A) Quantification of percent ZO1 colocalized with Occludin in the "ileum-like" region of CV and GF larvae. (B) Representative immunohistochemistry of CV WT and *sox10* mutant ZO1, Occludin and colocalization in "ileum-like" region. (C) Quantification of percent ZO1 colocalized with Occludin in the "colon-like" region of CV and GF larvae. (D) Representative immunohistochemistry of ZO1, Occludin and colocalization in "colon-like" region. (E) Quantification of percent ZO1 colocalized with Claudin-15 in the 'colon-like' region of CV and GF larvae. (F) Representative immunohistochemistry of ZO1, Claudin-15 and colocalization in "colon-like" region. Scale bar = 20μm. Each point in A, C and E is the average value of 2–3 sections per fish. Staining was performed on larvae from two independent GF derivations (n>7). Boxes represent the first to third quartiles, center bar denotes the median, and whiskers the maximum and minimum of each dataset. * p < 0.05. ANOVA followed by Tukey's post hoc test.

[51]. Zebrafish Claudin15, which is responsible for ion secretion, does not influence macro-molecule permeability [52], and does not undergo endocytosis. We examined colocalization of Claudin15 with ZO1 in the distal intestine and found that this colocalization did not differ between WTs and *sox10* mutants in either CV or GF conditions (Fig 3E and 3F). These data suggest that the intestinal hyperpermeability of CV and GF *sox10* mutants results specifically from disruption of the paracellular leak pathway through altered Occludin dissociation from ZO1 in the "colon-like" region.

### The ENS regulates luminal pH

We previously demonstrated that intestinal inflammation in *sox10* mutants results from microbial dysbiosis [9] and here we show that tight junction mediated intestinal hyperpermeability in *sox10* mutants precedes microbially induced inflammation. Thus, we next wanted to understand whether the intestinal environment of *sox10* mutants was altered in a way that promotes microbial dysbiosis. Tight junctions are known to play a role in intestinal ion balance and bidirectional fluid flow [53] and the ENS is known to regulate ion and pH balance through pancreatic and intestinal epithelial cell secretions [54–57]. Therefore, we hypothesized that luminal pH might be altered in *sox10* mutants. The larval zebrafish intestinal lumen is alkaline, with a typical pH value above 7.5 [58,59]. We assessed luminal pH of WT and *sox10* mutants using two different pH indicators, m-cresol purple (Fig 4A) and phenol red (Fig 4C) and quantified the color using the red, green, blue integer values [60] of these indicators within the zebrafish proximal intestine, commonly referred to as the bulb. We used color measurements of pH standards in zebrafish embryo medium (EM) as comparators for the values obtained in the zebrafish bulb, but distortions in color caused by imaging through zebrafish tissue precluded assigning specific pH values to the *in vivo* measurements. We gavaged each indicator into 6 dpf CV and GF WTs and *sox10* mutants and imaged the fish 20 minutes later, allowing time for the indicator to change color in its new environment. We found that CV *sox10* mutants had lower intestinal pH, as demonstrated by increased color integer values, compared to CV WTs (Fig 4B and 4D). To determine whether this lower pH could result from microbial

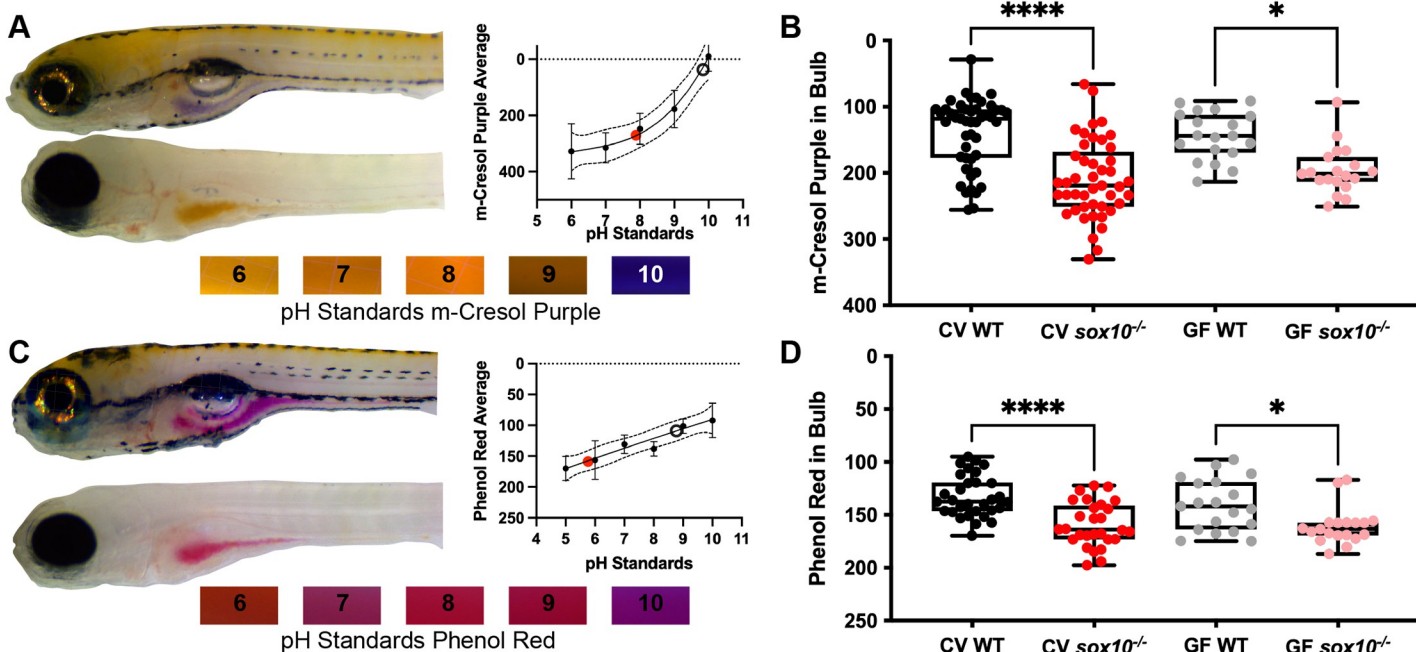

**Fig 4. *sox10* mutant intestinal lumens are more acidic than wild types.** (A) Top left, representative images of WT (top) and *sox10* mutant (bottom) larvae 20 min after m-cresol purple gavage. Bottom, pictures of indicator dyes in sterile embryo medium adjusted to known pHs; graph shows quantification of standards. (B) Quantification of luminal pH in intestinal bulb plotting red + green–blue integer values as described in the Methods. (C) Top left, representative images of WT (top) and *sox10* mutant (bottom) larvae 20 min after phenol red gavage. Bottom, pictures of sterile embryo medium adjusted to known pHs; graph shows quantification of standards. (D) Quantification of luminal pH in intestinal bulb plotting red integer values. Integer values correspond with red, green and blue channel pixel intensities between 0 and 255 identified with "RGB measure" Image J plugin (see Methods). (A&C) Each dot is an average of at least 3 replicates of pH standard values. Black circle indicates value of WT representative image and red dot indicates value of *sox10*−/− representative image. (B&D) Each dot is an individual fish; n>17 for each condition. Boxes represent the first to third quartiles, center bar denotes the median, and whiskers the maximum and minimum of each dataset. * p < 0.05, ****p<00001. ANOVA followed by Tukey's post hoc test.

physiologies or microbially induced inflammation, we derived *sox10* mutants and WT siblings GF and found that GF *sox10* mutants also had significantly lower pH, compared to GF WTs (Fig 4B and 4D). The relative pH differences between WT and *sox10* larvae were consistent between the two pH indicators. Interestingly, CV individuals had more variable pH values compared to GF individuals, and the average GF WT pH tended to be lower than that of CV WTs. These observations suggest that the microbiota can affect intestinal pH, but that these effects are subtle, and the higher variability among CV animals may be due to normal differences in the microbial communities of animals used in different experiments done on different days. These data indicate that luminal pH balance is independent of microbially induced inflammation in *sox10* mutants and could be a primary defect leading to microbial dysbiosis.

## Decreased luminal pH is necessary and sufficient for intestinal inflammation

Our results indicated that decreased intestinal transit, hyperpermeability, and decreased luminal pH all occur in *sox10* mutants independent of the microbiota and thus could be primary defects driving dysbiosis and hyperinflammation. We sought to test whether alterations in luminal pH could explain the dysbiosis-induced hyperinflammation seen in *sox10* mutant intestines by manipulating intestinal pH with pharmacological agents. Although zebrafish do not have dedicated acid-producing Parietal cells, we found evidence for zebrafish enterocyte expression of proton pump genes, including *atp6v0ca*, *atp6va2b*, and *atp6ap2*, within published intestinal tissue single cell transcriptome data [35,61,62]. We therefore reasoned that we could increase luminal pH using the common proton-pump inhibitor omeprazole. We exposed CV WTs and *sox10* mutants to omeprazole for 24 hours and measured intestinal pH and neutrophil numbers. Consistent with our prediction, we found that incubation in omeprazole increased the luminal pH of both CV WTs and CV *sox10* mutants (Fig 5A and 5C).

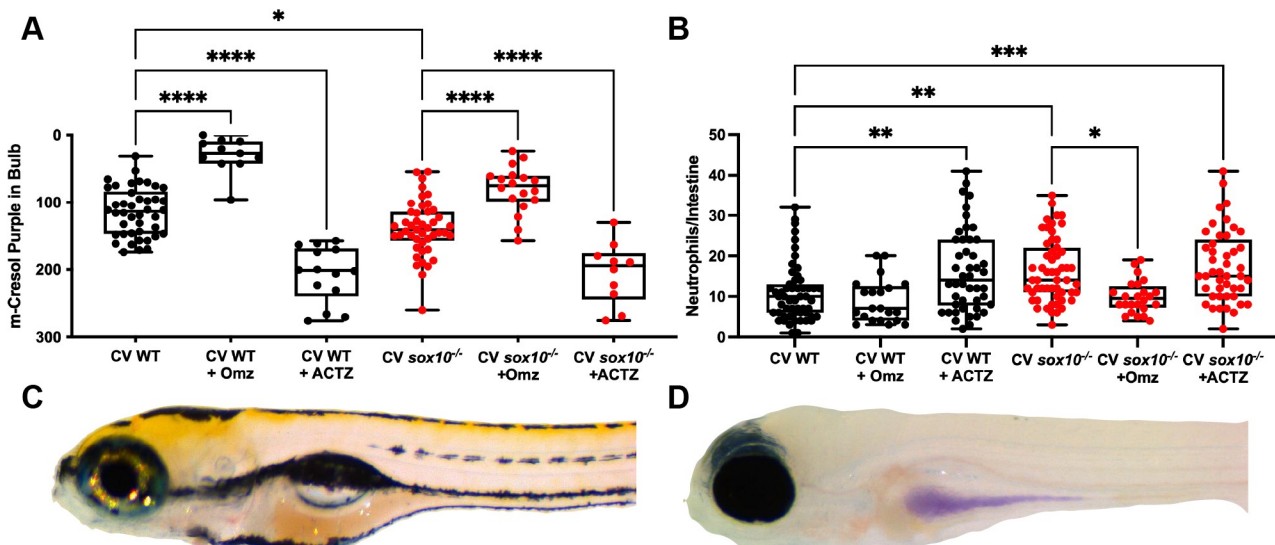

**Fig 5. Decreased pH is necessary and sufficient for intestinal hyperinflammation.** (A) Quantification of luminal pH in intestinal bulb plotting red + green–blue integer values (see Methods) of CV WT and *sox10* mutants after 24 hour exposure to 200 μM proton pump inhibitor omeprazole (Omz) or carbonic anhydrase inhibitor acetazolamide (ACTZ) 20 min after m-Cresol Purple gavage. (B) Quantification of intestinal neutrophil number per distal intestine in CV WT and *sox10* mutants after 24 hour exposure to 200 μM Omz or ACTZ. (C) Representative images of WT after ACTZ treatment and (D) *sox10* mutant after Omz treatment. In A & B, each dot is a fish, n>10. Boxes represent the first to third quartiles, center bar denotes the median, and whiskers the maximum and minimum of each dataset. * p < 0.05, **p<0.01, ****p<00001. ANOVA followed by Tukey's post hoc test.

Omeprazole also reduced the number of intestinal neutrophils in *sox10* mutants to WT levels, thus reversing the *sox10* hyperinflammation phenotype (Fig 5B). To validate that this reduced inflammation was due to increased luminal pH in *sox10* mutants, we treated CV WTs and *sox10* mutants with N,N-Dimethylamiloride (DMA), an inhibitor of the sodium proton exchanger [63]. We found that DMA also increased luminal pH and, and consistent with our results with omeprazole treatment, DMA also decreased intestinal neutrophils in *sox10* mutants (S2A and S2B Fig). These results provide strong evidence that the reduced luminal pH of *sox10* mutants induces the hyperinflammatory state.

To test whether decreased luminal pH is sufficient to induce a hyperinflammatory state, we exposed CV WTs and *sox10* mutants to the common carbonic anhydrase inhibitor acetazolamide (ACTZ) for 24 hours. We found that ACTZ incubation decreased luminal pH of both CV WTs and CV *sox10* mutants (Fig 5A and 5D). ACTZ incubation of WT larvae also increased intestinal neutrophils to *sox10* mutant levels (Fig 5B). These results provide strong evidence that reduced luminal pH is sufficient to induce a hyperinflammatory state.

### Luminal pH regulates abundance of proinflammatory intestinal *Vibrio*

To understand the mechanisms by which luminal pH modulates inflammation, we tested whether omeprazole and acetazolamide influence *Vibrio* abundance in larval zebrafish. We previously showed that *sox10* mutants have dysbiotic microbiota resulting from an overabundance of zebrafish-associated *Vibrio* strains [9]. We also previously showed that *Vibrio* strain ZWU0020 (*Vibrio* Z20) promotes intestinal inflammation in GF zebrafish [9,37,42]. *Vibrio* Z20 is phylogenetically closely related to the *Vibrio* strains found in increased abundance in *sox10* mutants [9]. To establish an assay for *Vibrio* Z20 colonization, we exposed larvae to fluorescently tagged *Vibrio* Z20-GFP [42] (hereafter referred to as *Vibrio*-GFP) for 24 hours and assessed *Vibrio*-GFP intestinal abundance by both fluorescence microscopy and dilution plating. As we have described previously, this strain of *Vibrio* localizes to the intestinal bulb (Fig 6A), because it is highly planktonic and resistant to intestinal motility [41,42], in contrast to more aggregated zebrafish commensal bacteria that are displaced more distally along the intestine [64]. We found that *sox10* mutants had higher levels of *Vibrio*-GFP relative to WT siblings, as quantified by intestinal GFP fluorescence (Fig 6D), and total number of GFP-positive colony-forming units (GFP-CFUs) per intestine (Fig 6C). We further established that intestinal GFP fluorescence was correlated with GFP-CFUs (Fig 6B). To address whether we could correct the *Vibrio* overgrowth in *sox10* mutants by increasing luminal pH, we added omeprazole at the same time as *Vibrio*-GFP inoculation and assessed abundance 24 hours later. Omeprazole reduced *Vibrio*-GFP abundance in both CV WTs and *sox10* mutants as shown by GFP-CFU (Fig 6C; CV WT vs CV WT+OMZ t-test p = 0.01) and fluorescence (Fig 6D; CV WT vs CV WT+OMZ Student's t-test p = <0.0001), but did not influence *Vibrio*-GFP CFU in the surrounding flask EM (Fig 6F). These results suggest that omeprazole treatment resulted in a specific change in the host environment, altering the ability of *Vibrio* to achieve high intestinal abundance. DMA treatment also decreased intestinal abundance of *Vibrio*-GFP (S2C and S2D Fig; CV WT vs CV WT+DMA t-test p = 0.0003), but additionally resulted in a decrease in *Vibrio*-GFP in the flask water (S2E Fig), suggesting DMA may decrease *Vibrio* abundance by changing both the host environment and by influencing *Vibrio* in the flask water through an unknown mechanism. To learn whether we could induce *Vibrio* overgrowth in WT zebrafish by decreasing luminal pH, we added acetazolamide at the same time as *Vibrio*-GFP inoculation and assessed abundance 24 hours later. Acetazolamide increased *Vibrio*-GFP abundance in CV WTs as measured by fluorescence (Fig 6D) and by dilution plating (Fig 6C; CV WT vs CV WT + ACZ t-test p = 0.003), but did not influence *Vibrio*-GFP CFU in the flask water (Fig

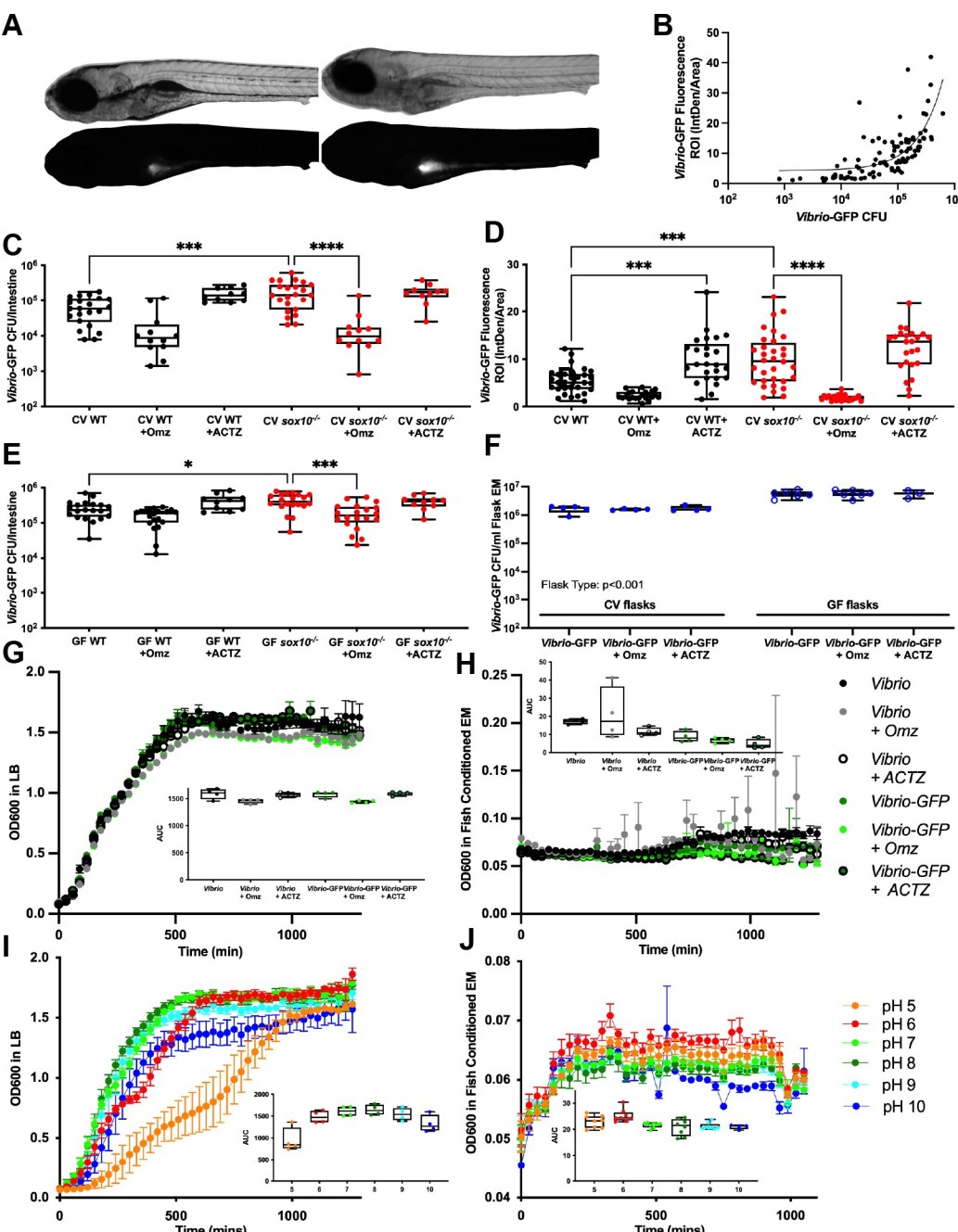

**Fig 6. Luminal pH regulates *Vibrio* abundance.** (A) Representative images of bright field (top) and fluorescence (bottom) showing WT (left) and *sox10*−/− (right) larvae after 24 hour exposure to *Vibrio*-GFP. (B) Fluorescence intensity of intestinal lumen correlates with abundance of *Vibrio*-GFP Colony Forming Units/intestine (CFU/intestine). (C) Quantification of *Vibrio*-GFP colonization level (CFU/intestine) in CV WT and *sox10* mutants after 24 hour exposure to *Vibrio*-GFP and 200 μM Omz or ACTZ (CV WT vs CV WT+OMZ t-test p = 0.01; CV WT vs CV WT + ACTZ: t-test p = 0.003). (D) Quantification of *Vibrio*-GFP luminal fluorescent intensity in CV WT and *sox10* mutants after 24 hour exposure to *Vibrio*-GFP and 200 μM Omz or ACTZ (CV WT vs CV WT+OMZ t-test p = <0.0001). (E) Quantification of *Vibrio*-GFP colonization level (CFU/intestine) in GF WT and *sox10* mutants after 24 hour exposure to *Vibrio*-GFP and 200 μM Omz or ACTZ (CV WT vs CV WT+OMZ t-test p = 0.01; CV WT vs CV WT + ACTZ: t-test p = 0.003). (F) Quantification of *Vibrio*-GFP colonization level (CFU/mL) in CV and GF flask embryo media (EM) after 24 hour exposure to 200 μM Omz or ACTZ. (G-H) *Vibrio* and *Vibrio*-GFP growth curves and area under the curve (AUC) in LB (G) and fish conditioned EM (H) with 200 μM Omz or ACTZ. (I-J) *Vibrio* growth curves and AUC in controlled pH LB (I) and fish conditioned EM (J). Each dot is a fish; n>12 for each condition (B-E), each dot is a flask (F), each dot is an average absorbance of 4 experimental replicates of at least 2 individual experiments (G&H). Boxes represent the first to third quartiles, center bar

denotes the median, and whiskers the maximum and minimum of each dataset. * p < 0.05, **p<0.01, ***p<0.001, ****p<00001. One-way ANOVA followed by Tukey's post hoc test (C-E). Two-way ANOVA followed by Šídák multiple comparisons test (F).

6F), suggesting that acetazolamide treatment resulted in a change in the host luminal environment, increasing the ability of *Vibrio* to achieve high intestinal abundance.

We next tested whether the impacts of these pharmacological manipulations of pH on *Vibrio* abundance were direct or mediated through other members of the microbiota. We performed the same manipulations on WT and *sox10* mutant zebrafish mono-associated with *Vibrio-GFP*. We observed strikingly similar patterns of *Vibrio* abundance in mono-associations as in the context of a CV microbiota. *Vibrio* abundance was increased in GF *sox10* larvae compared to GF WT, decreased in *sox10* animals with omeprazole, and increased in WTs with acetazolamide (Fig 6E; GF WT vs GF WT+ACTZ t-test p = 0.027). As expected, *Vibrio* abundance tended to be higher across all mono-associated conditions and flasks in the absence of competition from other microbes, and neither the omeprazole nor acetazolamide treatments impacted *Vibrio* abundance in the water of mono-associated flasks (Fig 6F).

To confirm that the pharmacological treatments did not directly influence *Vibrio*-GFP, we performed *in vitro* bacterial growth and swimming assays. We found that addition of omeprazole, acetazolamide, or DMA to either nutrient-rich lysogeny broth (LB; Figs 6G and S2F) or nutrient-limited fish conditioned EM [65] (Figs 6H and S2G) did not appreciably influence *Vibrio* Z20 or *Vibrio*-GFP growth, as measured by the area under the curve, or swimming radius on soft agar plates (S2H and S2I Fig). Taken together, these data indicate that a more acidic intestinal lumen is both necessary and sufficient to allow *Vibrio* to expand within the microbiota, suggesting that the microbial dysbiosis we previously reported in *sox10* mutants is a consequence of the lower luminal pH in *sox10* mutants compared to WTs.

*Vibrio* Z20 is closely related to human disease-associated *Vibrio cholerae* which are known to undergo an acid tolerance response that promotes survival in more acidic environments [66]. We tested whether *Vibrio* Z20 had altered growth in either LB or fish-conditioned EM across a range of pH values between 5–10. In rich medium, *Vibrio* Z20 achieved highest stationary state levels at pH 6–7, as indicated by the area under the curve, with a slight growth delay in pH 6 conditions (Fig 6I). A more pronounced growth delay was observed at pH 5 and reduced stationary state levels were seen in both pH 5 and 10 conditions. No growth was observed at pH 4 or 11. *Vibrio* Z20 and *Vibrio*-GFP showed minimal growth in fish-conditioned EM, but with similar trends of highest abundances observed in slightly acidic conditions of pH 5–6, and lower abundances at pH 8–10, as indicated by the area under the curve (Fig 6J). These data support our *in vivo* observations that *Vibrio* Z20 has a growth advantage in the mildly acidic intestinal lumens of *sox10* mutants compared to the more basic intestinal lumens of WTs.

## Discussion

Dysbiotic microbiota cause or propagate many inflammatory gastrointestinal diseases, but the factors that initiate dysbiosis have been difficult to uncover because of extensive feedback among the cellular players that contributes to intestinal environment complexity [28,67,68]. Host mechanisms that modulate environmental features relevant to microbial growth have been labeled "habitat filters;" alterations to these filters can select for pathobionts and lead to microbial dysbiosis [69]. Inflammation alters the chemical and physical environment of the intestine in ways that pro-inflammatory bacteria can exploit, creating a self-propagating inflammatory state. In some instances, the initial trigger is an infectious microbe, such as *Clostridium difficile*, which can cause severe and persistent inflammation and dysbiosis [70]. In

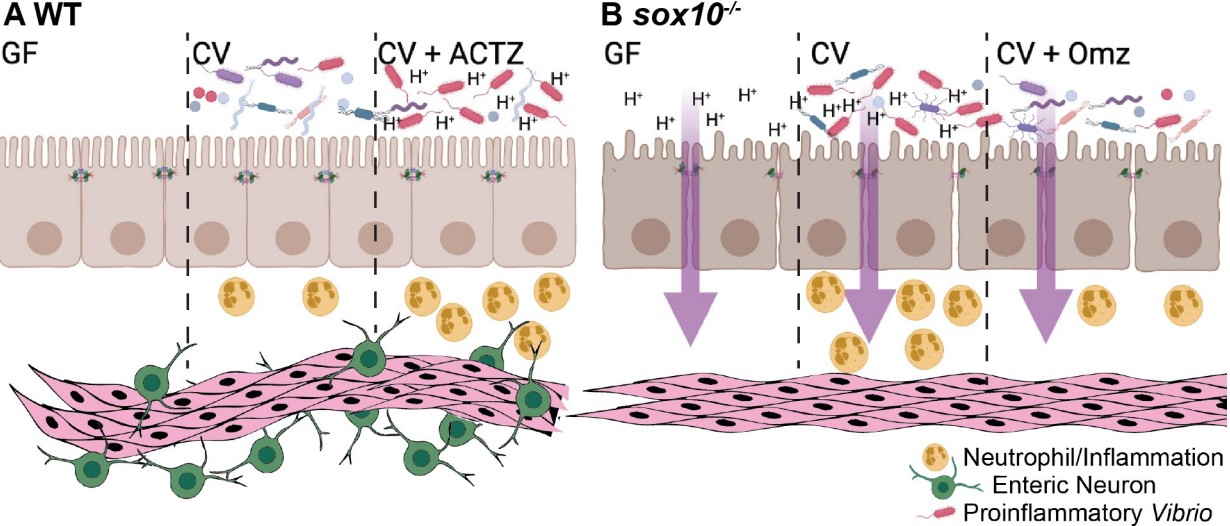

**Fig 7. Proposed model of luminal pH driven intestinal pathology.** (A) WT larvae with an ENS maintain luminal pH in GF and CV conditions. WT CV larvae maintain a healthy balance of microbiota and surveying neutrophils. ACTZ treatment of CV WT larvae decreases luminal pH, driving increased proinflammatory *Vibrio* abundance and neutrophil influx. (B) *sox10* mutants lacking the ENS have decreased intestinal transit, hyperpermeability, and decreased luminal pH in both GF and CV conditions. Decreased luminal pH promotes an increase in proinflammatory *Vibrio* leading to increased neutrophils and a hyper-inflammatory state. Omz treatment to CV *sox10* mutants increases luminal pH, resulting in decreased *Vibrio* abundance and ameliorated hyperinflammation. Model created with BioRender.com.

other cases, the initiation of dysbiosis is associated with host genetic predisposition [71], but the responsible cell types and molecular mechanisms have been difficult to determine. Analyses of the genetic basis of intestinal inflammatory disease have focused mostly on intestinal immune and epithelial cells [72], but the ENS is also an important cell population to consider [73–75]. We previously showed that loss of the ENS in zebrafish, due to mutation of the gene encoding the Sox10 transcription factor, results in spontaneous intestinal inflammation caused by dysbiotic expansion of pro-inflammatory *Vibrio* strains [9]. Here we used gnotobiology to uncover the primary tissue defects that create an intestinal environment permissive for dysbiosis. Our data show that loss of the ENS results in a more acidic intestinal lumen, which promotes expansion of pro-inflammatory *Vibrio* and induces intestinal inflammation (Fig 7).

## The ENS regulates the intestinal lumen pH

Our finding that *sox10* deficient intestines are more acidic than WTs in both CV and GF conditions highlights the importance of the ENS in maintaining the chemical environment of the intestinal lumen independent of microbial impacts. ENS function is important for regulating intestinal epithelial barrier function [76], ion transport across intestinal epithelial cells [77], and luminal pH [54–56]. The ENS has been suggested to influence tight junction protein expression and intestinal permeability [31,32], yet which specific components of the ENS and in which contexts is debated [78,79]. Here, we used gnotobiology to uncouple barrier function and inflammation. We found that CV *sox10* mutants exhibit hyperpermeability prior to the onset of intestinal inflammation and that GF *sox10* mutants exhibit hyperpermeability even in the absence of inflammation. In addition, we show that the paracellular leak pathway is disrupted in *sox10* distal intestines even in the absence of microbiota and inflammation. Interestingly, alterations in tight junction integrity occurred in the "colon-like" region, but not the "ileum-like" region, which coincides with our observation that we tend to see more neutrophils in the distal intestine. We hypothesize that the decreased barrier function in the "colon-

like" region contributes to the increased neutrophil influx in this region. Although *Vibrio* Z20 is distributed predominantly toward the anterior intestine [41,64], where it elicits a strong tumor necrosis factor (Tnf) cytokine and macrophage response [42], it is continually flushed out of the intestine and its proinflammatory products may be sensed especially acutely in the more permeable *sox10* distal intestine. Many enteric neuropathic disorders and other systemic neurodegenerative diseases with altered ENS function have impaired intestinal barrier function, yet whether this is a direct result of ENS malfunction or secondary to its direct influence on immune, endothelial, or smooth muscle cells is unclear [80]. Interestingly, intestinal bowel disease (IBD), and specifically individuals with Crohn's Disease, exhibit intestinal hyperpermeability [24,81], a defect that is being targeted for treatment to complement current anti-inflammatory therapies [82,83]. *In vitro* and *ex vivo* studies suggest that the ENS reduces hyperpermeability by releasing the neurotransmitter vasoactive intestinal peptide (VIP) [31,32,84], however the positive feedback loop of inflammation and hyperpermeability makes it difficult to establish the mechanism of action *in vivo* [85,86]. Collectively our results demonstrate that maintenance of intestinal barrier function is a primary function of the ENS. We did not investigate a mechanism by which the hyperpermeability of *sox10* mutant zebrafish could contribute to increased intestinal lumen acidity; however, a reasonable hypothesis is that increasing flow of macromolecules and water across the intestinal epithelium could result in increased luminal concentration of protons or decreased concentration of buffering molecules.

The ENS also regulates the chemical environment of the intestinal lumen through its influence on secretion from epithelial cells, the gallbladder, and the pancreas, all of which can alter intestinal luminal pH [54–57]. Although we did not directly investigate these processes, they could be altered in *sox10* deficient zebrafish. Intestinal enteroendocrine cells sense decreased pH and release paracrine signals to the ENS which directly control the flow of bile and pancreatic juices [57] and induce intestinal epithelial bicarbonate secretion [87–90]. The ENS innervates the zebrafish intestine and pancreas in early larval stages, a process that fails to occur in *sox10* mutants [7,91]. Zebrafish bile consists primarily of $C_{27}$ bile alcohol and $C_{24}$ bile acid which are released after lipid ingestion and could decrease luminal pH [92]. We show that *sox10* mutants have a less alkaline intestinal lumen than their WT siblings independent of the microbiota. It is possible that ENS dependent bicarbonate secretions from epithelial cells and regulation of basic pancreatic secretions make the zebrafish intestinal lumen alkaline, which could contribute to the more acidic intestinal lumens of *sox10* mutants lacking an ENS. Our characterization of the GF *sox10* deficient intestine demonstrates that alteration in the chemical environment of the intestinal lumen in these mutants is a primary defect that could select for an altered microbiota.

## The chemical and physical environment of the intestine shapes the microbiota

In conditions of uncontrolled intestinal inflammation, populations of certain opportunistic pathogens expand because of their ability to thrive in the altered habitat [93]. Our study of the intestinal environment of *sox10* mutant larval zebrafish identifies alkaline luminal pH as a critical habitat filter that normally constrains the expansion of proinflammatory *Vibrio* strains. We showed that *Vibrio* Z20 has a growth advantage *in vitro* in mildly acidic versus alkaline pH media. Human isolates of *Vibrio cholera* deploy an acid tolerance response to survive the low pH of the stomach and successfully colonize the small intestine [94]. Whether *Vibrio* Z20 uses a similar acid tolerance program for its competitive success in the *sox10* deficient intestine remains to be determined.

Intestinal motility is another habit filter that constrains microbial composition. We showed previously that in zebrafish mutants lacking the ENS due to mutation in the *ret* gene, a bacterial strain is able to persist that would normally be outcompeted in the presence of intestinal flow forces [41]. Coordinated, ENS-regulated intestinal contractions begin in larval zebrafish at 4 dpf and are robustly established by 6 dpf [44,95]. Here we assayed intestinal transit as a functional readout of these contractions and showed that *sox10* mutants have decreased intestinal transit even in the absence of their microbiota, identifying another primary defect in these mutants. Unlike the barrier dysfunction, however, which precedes intestinal inflammation, the reduced transit in *sox10* mutants is apparent at a similar timepoint as the increased neutrophils. We therefore suspect that in the *sox10* mutant intestine, the impaired barrier and resulting altered luminal chemistry are more important triggers of dysbiosis than the transit defects, although the slower transit, especially in the context of increased inflammation, may contribute to selecting for and perpetuating a pro-inflammatory intestinal microbial community.

## Manipulating the luminal environment to mitigate dysbiosis-associated diseases

As a model of HAEC and an example of spontaneous intestinal inflammation, *sox10* deficient zebrafish provide potential new insights into the drivers of dysbiosis in human inflammatory intestinal disease. Although pH changes have not been described in HAEC, decreased luminal pH has been reported in IBD patients [96,97]. More generally, many gastrointestinal disorders are associated with altered expression of ion channel genes including the sodium potassium pump that directly regulates intestinal pH [98,99]. These observations point to luminal pH as an important habitat filter that may be compromised in inflammatory GI disorders. There is a pressing need for treatment strategies for HAEC, which reoccurs in 25% of patients post-surgical removal of the aganglionic colon and is the leading cause of mortality [33,100].

We utilized the commonly prescribed proton pump inhibitor (PPI) omeprazole [101,102] to neutralize the luminal environment of the intestine and showed that this was sufficient to constrain proinflammatory *Vibrio* colonization and prevent intestinal inflammation. Omeprazole is prescribed to patients to treat GERD or acid reflux [103] because it directly increases stomach pH, yet the influence of PPIs on commensal microbial communities in the lower GI tract is less well studied [104,105]. Utilizing omeprazole to correct proinflammatory dysbiosis may be useful for combatting inflammation. Notably, omeprazole treatment of cystic fibrosis transmembrane conductance regulator (CFTR) knock out mice, which have decreased bicarbonate ion secretion, more acidic intestinal lumens, and proinflammatory dysbiosis [106], results in decreased neutrophil abundance in the pancreas [107]. Our result specifically links omeprazole's influence on luminal pH to reduce proinflammatory bacterial selection, resulting in reducing intestinal neutrophils.

We used the commonly prescribed carbonic anhydrase inhibitor acetazolamide [108] to manipulate the luminal environment in the opposite direction and showed that this allowed expansion of proinflammatory *Vibrio* in the wild type intestine. Acetazolamide is prescribed to patients to treat epilepsy, glaucoma, edema, altitude sickness, and many off label applications [109]. As a carbonic anhydrase inhibitor, it prevents the breakdown of carbonic acid into bicarbonate and hydrogen ions, thus interfering with a process of acid neutralization. Because carbonic anhydrase is found in the intestinal mucosa, red blood cells, and renal tubes, acetazolamide decreases pH of both the intestinal lumen and blood. Recently acetazolamide has been used to reduce intestinal polyps [110], which have been linked to colonic dysbiosis [111]. Collectively our experiments with omeprazole and acetazolamide demonstrate that intestinal pH

is a microbiota habit filter which can be manipulated for therapeutic treatment of intestinal inflammation.

## Conclusion

We utilized zebrafish as a powerful gnotobiotic model to dissect the complex relationship between the ENS, epithelial barrier, inflammation, and the microbiota. We demonstrated the critical role played by the ENS in shaping the intestinal luminal environment and restricting selection of a pathobiont. Our analysis reveals how, without this constraint, proinflammatory bacteria thrive, driving intestinal pathology. Furthermore, we have demonstrated that commonly prescribed pharmacological agents omeprazole and acetazolamide can be used to manipulate luminal pH, resulting in predictable alterations in bacterial composition and inflammation. Our work demonstrates luminal pH to be an important and therapeutically malleable habitat filter for the intestinal microbiota.

## Methods

### Ethics statement

All experiments were conducted according to protocols approved by the University of Oregon Institutional Animal Care and Use Committee (protocol numbers 15–15, 18–29 and 20–16) and followed standard zebrafish protocols [112].

### Zebrafish husbandry

Heterozygous *sox10^t3* (referred to as *sox10^-/+)* fish were maintained as described at 28˚C [112]. Homozygous *sox10* mutants were obtained by mating heterozygotes and identified by lack of pigmentation [8]. No defects were observed in heterozygous siblings, which have pigment, develop normally and survive to adulthood, and thus they were grouped with homozygous WTs [44]. For all experiments, WT siblings, heterozygous, and homozygous *sox10* mutants were cohoused and inoculated with 2% parental fish tank water in the embryo medium (EM).

### Gnotobiological fish husbandry

Zebrafish embryos were derived GF as previously described [113–115] and verified by plating the media and counting colony forming units (CFUs) and by visual inspection using a compound microscope. Briefly, fertilized eggs were placed in antibiotics for five hours, washed with sterile EM, incubated in PVP-1 and bleach, and then placed in flasks filled with sterile EM. At 3 dpf, to aid larval hatching from the chorion, embryos were washed over a 40 μm filter with sterile EM and placed in a new flask containing sterile EM. At 6 dpf, flasks were inspected on a Leica DMi1 inverted microscope. Flasks with identifiable bacteria and/or high amounts of debri were discarded. CV or GF fish were inoculated with fluorescently tagged *Vibrio*-GFP ($10^6$ bacterial cells/ml) as previously described [9,42] at 5 dpf. Analysis was performed after 24 hours.

### Histological analysis, quantification of neutrophils

To quantify neutrophils, 6 dpf zebrafish larvae were fixed in 4% paraformaldehyde in PBS overnight at 4˚C. Whole larvae were stained with LEUCOGNOST POX reagents (VWR EM1.16303.0002) following the manufacturer's protocol and processed and analyzed as previously described [113]. Briefly, the intestine was dissected from the larvae and neutrophils were counted in the mid and distal intestine.

## Intestinal permeability, transit measurements

Oral microgavage of 4, 5, 6, and 7 dpf larvae was performed as previous described [30,45] with the following modifications. Briefly, larvae were anesthetized with Tricaine (168 mg/L; Western Chemical, Inc., Ferndale, WA), gavaged with 4.6 nl of a 1% solution of fluorescein isothiocyanate-conjugated (FITC)-dextran (FD4; 4 kD; Sigma) or tetramethylrhodamine isothiocyanate (TRITC)-dextran (TD4; 4 kD; Sigma) and 2% phenol red solution. The most distal location of phenol red in the intestine was identified immediately following gavage and at time of imaging based on the somite to which it was adjacent [116]. Larvae were mounted in 4% (wt/vol) methylcellulose (Fisher, Fiar Lawn, NJ) and imaged 30 min after gavage using a Leica TCS SPE widefield microscope, with 200 ms exposure time. Image J software (NIH) was used to measure the integrated fluorescence density in a 4 somite length region of interest (ROI) over the aorta. A background ROI of the same size was measured over the notochord and subtracted from the aorta ROI. Fluorescent measurements were normalized to CV WT for each experiment.

## Immunohistochemistry

6 dpf larvae were fixed in 2% TCA in PBS for 4 hrs, washed with PBS, and stored in the freezer overnight. Larvae were cryopreserved and sectioned at 16 µm by University of Oregon Institute of Neuroscience Histology Facility. Sections were stained with mouse anti-ZO-1 IgG1 monoclonal antibody (Thermo Fisher, cat #339100, 1:350), rabbit anti-Occludin IgG polyclonal antibody (Thermo Fisher, cat #71–1500, 1:200), and rabbit anti-Claudin 10 IgG polyclonal antibody (Thermo Fisher, cat# 38–8400, 1:200, which detects zebrafish Claudin 15 [52]). Counting sections from the vent, regionally similar sections were imaged on a Leica TCS SPE confocal fluorescence microscope. For each region, three representative sections per fish were analyzed using Imaris software (Oxford Instruments, version 9.5.1). Apical ZO1 expression was isolated by creating a manual surface in Imaris and then ZO1:Occludin or ZO1:Claudin colocalization was measured.

## Luminal pH measurements

Oral microgavage of 6 dpf larval zebrafish was performed using pH indicators m-cresol purple (0.2%; Thermo-Fisher, 2303-01-7) or phenol red (2%; Sigma, 143-74-8). The fish were then imaged on a Leica MZ10F stereoscope using a Leica MC190 HD color camera. EM solutions of known pHs were imaged within a pipet tip in the same environment as the fish. The color of the bulb and pH standards were determined using the Image J software plugin "RGB measure." M-cresol purple values are presented by adding the red and green channel values and subtracting the blue channel value. The phenol red values are presented as the red channel value.

## Pharmacological treatment

Omeprazole (200 µM, Sigma, cat #0104), 5-(N,N-Dimethyl)amiloride hydrochloride (100 µM, Sigma, cat #A4562) Acetazolamide (200 µM, Sigma, cat #6011) were added to 5 dpf zebrafish larvae and pH, neutrophil, and *Vibrio*-GFP abundance measurements taken 24 hours later.

## *In vivo Vibrio* quantification

Zebrafish larvae were humanely euthanized with 300 mg/L Tricaine at 6 dpf, mounted in 4% methylcellulose, imaged on a Leica MZ10F stereoscope fluorescent microscope, and their intestines dissected using sterile technique. Dissected intestines were placed in sterile EM,

homogenized, diluted and cultured on tryptic soy agar plates (TSA; BD, Sparks MD). Flask water was also diluted and cultured on TSA plates. After incubation at 32˚C for 24 hours, fluorescent colonies were counted. Fluorescent images of zebrafish were analyzed using Image J software measuring the integrated density of the intestine (region of interest (ROI)). The integrated density of background fluorescence was subtracted by the intestinal fluorescence and divided by the ROI area.

### *In vitro Vibrio* growth measurements

Fish conditioned EM was prepared using methods established in [65]; briefly, embryos were derived GF, and collected into flasks with sterile EM. At 5–6 dpf, media was separated from larvae and sterilized with a 0.2μm filter. The pH of Lysogeny Broth (LB) and fish conditioned EM was adjusted using HCl or NaOH.

Bacterial strains were maintained in 25% glycerol at −80˚C. Bacteria were grown overnight in 5 mL LB (10 g/liter NaCl, 5 g/L yeast extract, 12 g/L tryptone, 1 g/L glucose) at 30˚C with shaking. After the initial overnight growth in LB, cultures were diluted 1:100 into fresh LB or fish conditioned EM at varying pHs in equal volumes as treatments (Omeprazole, N, N-Dimethylamiloride, Acetazolamide, PBS control). The subcultured cell solutions were dispensed in triplicate or quadruplicate (i.e., 3–4 technical replicates; 200μl/ well) into a sterile 96-well clear flat bottom tissue culture-treated microplate. Absorbance measurements at 600 nm were recorded every 30 minutes for 20 hours (or until stationary phase) at 30˚C with shaking. Growth measurements were repeated at least 2 independent times (i.e., 2 biological replicates) with consistent results. *In vitro* growth of bacterial strains was assessed using a FLUOstar Omega microplate reader (BMG LABTECH, Offenburg, Germany).

### Statistics

Statistical analysis was performed using Graphpad Prism9 software. Statistical significance was defined as $p < 0.05$. Significance was determined using ANOVA followed by Tukey's pairwise comparisons unless otherwise noted. Throughout, boxplots represent the median and inter-quartile range; whiskers represent the max and min. Student's t-tests were performed on select comparisons and presented within the text. Permeability data were not normally distributed and bounded by 0, thus significance was determined using a Kolmogorov-Smirnov nonparametric t-test. For analysis of neutrophil number and intestinal transit over time and between genotypes (Fig 2), and analysis of *Vibrio* CFU in flask water (Fig 6F), significance was determined using a 2-way ANOVA followed by Šídák multiple comparisons test.

### Supporting information

**S1 Fig. Slower transit is correlated with hyperpermeability and increased intestinal neutrophils in WTs, but not *sox10* mutants.** Correlation analysis of (A) WT and (B) *sox10* mutant transit and neutrophils. Correlation analysis of (C) WT and (D) *sox10* mutant permeability and neutrophils. Correlation analysis of (E) WT and (F) *sox10* mutant transit and permeability. Each dot is a fish; n>18 for all conditions. Simple linear regression analysis on CV WT, GF WT, CV *sox10*, and GF *sox10* data points.
(TIF)

**S2 Fig. Sodium proton pump inhibitor N,N-Dimethylamiloride (DMA) decreases *Vibrio* abundance in zebrafish intestine and surrounding EM.** (A) Quantification of luminal pH in intestinal bulb plotting the value of red + green–blue integer values (see Methods) of CV WT and *sox10* mutants after 24 hour exposure to 100 μM N,N-Dimethylamiloride (DMA) 20 min

after m-Cresol Purple gavage. (B) Quantification of intestinal neutrophil number per distal intestine in CV WT and *sox10* mutants after 24 hour exposure to 100 μM DMA. (C) Quantification of *Vibrio*-GFP colonization level (Colony Forming Units/intestine) in CV WT and *sox10* mutants after 24 hour exposure to *Vibrio*-GFP and 100 μM DMA (CV WT vs CV WT +DMA t-test p = 0.0003). (D) Quantification of *Vibrio*-GFP luminal fluorescent intensity in CV WT and *sox10* mutants after 24 hour exposure to *Vibrio*-GFP and 100 μM DMA. (E) Quantification of *Vibrio*-GFP colonization level (CFU/ml) in surrounding EM after 24 hour exposure to 100 μM DMA. (F) Quantification of *Vibrio* and *Vibrio*-GFP growth curves in LB and (G) fish conditioned EM with 100 μM DMA. (H) Quantification of *Vibrio* and *Vibrio*-GFP swimming diameter in agarose with 100 μM DMA, 200 μM Omz, or 200 μM ACTZ. (I) Representative image of *Vibrio* and *Vibrio*+Omz growth diameters in agarose. Each dot is a fish; n>11 (B-D), each dot is a flask (E), each dot is an average absorbance of 4 experimental replicates (F&G). Boxes represent the first to third quartiles, center bar denotes the median, and whiskers the maximum and minimum of each dataset. $^*$ p < 0.05, $^{**}$p<0.01, $^{***}$p<0.001, $^{****}$p<00001. ANOVA followed by Tukey's post hoc test.
(TIF)

**S1 Data File. Statistical Tests.** Statistical tests for comparisons between all treatment groups. Each table includes statistics for individual panels, each figure separated by rows. P values presented and determined significant when p<0.05.
(XLSX)

**S2 Data File. Data.** Raw data for all presented graphs. Each figure separated by tab.
(XLSX)

## Acknowledgments

The authors would like to thank Adam Fries and the University of Oregon Genomics, Cell Characterization and Imaging Core Facility for expertise in confocal imaging and IMARIS software and Ellie Melancon and the UO Zebrafish Facility staff for fish husbandry.

## Author Contributions

**Conceptualization:** M. Kristina Hamilton, Karen Guillemin, Judith S. Eisen.

**Data curation:** M. Kristina Hamilton, Elena S. Wall.

**Formal analysis:** M. Kristina Hamilton.

**Funding acquisition:** M. Kristina Hamilton, Karen Guillemin, Judith S. Eisen.

**Investigation:** M. Kristina Hamilton, Elena S. Wall, Catherine D. Robinson.

**Methodology:** M. Kristina Hamilton, Elena S. Wall, Karen Guillemin, Judith S. Eisen.

**Resources:** M. Kristina Hamilton, Karen Guillemin, Judith S. Eisen.

**Supervision:** M. Kristina Hamilton, Karen Guillemin, Judith S. Eisen.

**Visualization:** M. Kristina Hamilton.

**Writing – original draft:** M. Kristina Hamilton.

**Writing – review & editing:** M. Kristina Hamilton, Catherine D. Robinson, Karen Guillemin, Judith S. Eisen.

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
