## [Decision Letter · Decision Letter 0]

29 Oct 2021

Dear Dr. Eisen,

Thank you very much for submitting your manuscript "Enteric nervous system modulation of luminal pH modifies the microbial environment to promote intestinal health" for consideration at PLOS Pathogens. As with all papers reviewed by the journal, your manuscript was reviewed by members of the editorial board and by several independent reviewers. In light of the reviews (below this email), we would like to invite the resubmission of a significantly-revised version that takes into account the reviewers' comments.

We cannot make any decision about publication until we have seen the revised manuscript and your response to the reviewers' comments. Your revised manuscript is also likely to be sent to reviewers for further evaluation.

Sincerely,

Andreas J Baumler

Associate Editor

PLOS Pathogens

Nina Salama

Section Editor

PLOS Pathogens

Kasturi Haldar

Editor-in-Chief

PLOS Pathogens

orcid.org/0000-0001-5065-158X

Michael Malim

Editor-in-Chief

PLOS Pathogens

orcid.org/0000-0002-7699-2064

Reviewer's Responses to Questions

**Part I - Summary**

Reviewer #1: Host-microbe interactions are of the utmost importance when it comes to pursuing a holistic understanding of the many factors that affect the health of macroorganisms. In humans and all other animals studied to date, these interactions with microbes are particularly impactful in the gastrointestinal tract. A “healthy” relationship between host and microbiota generally supports host immune function, colonization resistance against invading pathogens, host and microbe nutrition, and many other aspects of host function. When a dysbiotic state is achieved in the GI tract, often roughly defined by a pro-inflammatory shift in the composition or function of the microbiota, the host tends to suffer the consequences in the form of inflammation and a variety of other negative effects. Much work has been done in recent years with the overarching goal of achieving an integrated understanding of host-microbe and microbe-microbe interactions in the gut. While these efforts have uncovered many important interactions between the host and the microbiota, the role of the enteric nervous system in this complex environment has remained a particularly difficult research topic.

In this study, Hamilton et al. investigate the role of the ENS in maintaining homeostasis in the gut by turning to the larval zebrafish model. By harnessing a sox10 mutant, which lacks an ENS and develops pro-inflammatory dysbiosis, the authors were able to leap-frog from their previous observations in this model in order to gain a more mechanistic understanding of how the ENS regulates the microbial composition of the GI tract. The authors find that the lack of an ENS leads to hyperpermeability of the intestinal epithelium, slower gut transit times, and decreased luminal pH. Importantly, these effects of sox10 mutation were found to be independent of the microbiota.

I found this manuscript to be well-written and the experiments to be elegantly done.

Reviewer #2: In the manuscript titled “Enteric nervous system modulation of luminal pH modifies the microbial environment to promote intestinal health,” Hamilton et al. characterized the roles of the enteric nervous system (ENS) in intestinal homeostasis. The authors used the sox10 KO line, which was previously shown to lack ENS, to explore the alterations of intestinal physiology in zebrafish. They showed that sox10 deficient fish had altered intestinal transit, intestinal permeability, and luminal pH regulation. They provided evidence to support their claim that these alterations are independent of the presence of the gut microbiota, and therefore these alterations precede and therefore are causal to the dysbiosis and gut inflammation observed in the sox10 mutant. Hamilton et al. further showed that changes in the luminal pH are likely the cause of the overgrowth of proinflammatory Vibrio species in the sox10 mutant. The authors concluded that a primary function of the ENS is to regulate luminal pH, which plays a critical role in shaping the resident microbial community and regulating intestinal inflammation. This manuscript is well written, the hypotheses were clearly stated, and the evidence provided is largely in support of the author’s claims. However, the interpretation of some key experiments is problematic and will need to be revised what was observed experimentally. Please see specific comments below:

Reviewer #3: In this study, Hamilton et al utilize a zebrafish model lacking an enteric nervous system (ENS) due to a mutation in the sox10 gene to continue to investigate how the ENS influences the intestinal environment to modulate the gut microbiota. The authors had previously found that the ENS alters the composition of the microbiota, neutrophil recruitment in the gut lumen, and overgrowth of pro-inflammatory Vibrio strains. The authors utilize wild-type zebrafish with a conventional microbiota, wild-type germ-free zebrafish, sox10 mutant zebrafish with a conventional microbiota, and germ-zebrafish sox10 mutant zebrafish and report that intestinal transit, intestinal permeability, and luminal pH regulation are aberrant in sox10 mutants independent of the microbiota. Finally, they find that loss of sox10 leads to a drop in luminal pH and overgrowth of proinflammatory Vibrio strains. The data are incremental but interesting, and the findings have potential to have broad implications on the role of sox10 on pH and growth of proinflammatory members of the microbiota. My major criticism is that a lot of the data is variable and some of the differences reported to make major conclusions are not convincing. The authors would strengthen their conclusions if they repeated some of the experiments to increase the significance and confidence in the data. The manuscript would also benefit from improving the clarity of the writing. My comments are outlined below:

**Part II – Major Issues: Key Experiments Required for Acceptance**

Reviewer #1: I do not believe that this manuscript requires any major revisions.

Reviewer #2: Major issues:

1. Fig. 1D: While the sox10 mutant still exhibited decreased transit time compared to the CV WT, there was no difference between GF sox10 and WT, therefore the decrease was not due to host genotype per se but the microbiota status. Thus, the authors claim “decreased transit in sox10 mutants still occurred in the GF state” need to be revised to better reflect experiment results.

2. While the FD4 experiment showed that the increased intestinal permeability associated with sox10 was not microbiota dependent, the TD4 results suggested increased intestinal permeability was associated with microbiota status (Fig. 1E&F). Given the noted variability of this phenotype in the sox10 mutant, was the discrepancy a result of undersampling, as the FD4 experiment clearly had more data points than the TD4 experiment?

3. While it is plausible that the transmission of microbes was responsible for the comparable neutrophil numbers between WT and the sox10 mutant (Fig 2A), this claim can be better substantiated by doing similar experiments in separately housed animals.

4. While the authors showed good evidence that luminal pH correlated with Vibrio abundance, it is unclear whether luminal pH directly regulates Vibrio abundance or indirectly through influencing other competing microbes in the microbiota, or the combination of the two. This hypothesis can be easily tested using GF animals moncolonized with Z20.

Reviewer #3: -One major concern is in regard to the conclusions based on the statistical differences reported in some of the figures. The author’s use statistical analysis to determine whether phenotypes are dependent on sox10 or the microbiota. However, using statistics alone to make conclusions can be misleading especially when there is there is a quite a bit of variability and overlap in the data between groups. It’s important to also reply on common sense and judgement when interpreting the data and making conclusions. In some cases, the differences don’t appear very different but the authors report that they are statistically different, and in some cases, the data appear different between groups but the authors report that they are not statistically different. For example, in Figure 1D, the authors report that the difference in Transit between GF WT and GF sox10 -/- are not statistically different but it’s clear that there is a clear trend for reduced transit time in GF sox10 -/-. In fact, in GF sox10 -/- six animals had a transit time below the limit of detection, while GF WT had zero animals with this phenotype. In contrast, in Figure 1E the authors report that there is a statistically different (**p<0.01) difference in intestinal permeability between GF WT and GF sox10 -/- although there is also a clear trend, the difference doesn’t appear very different between these groups, with exception to the handful of GF sox10 -/- animals with high intestinal permeability. It’s not clear why in Figure 1E, the difference between GF WT and GF sox10 -/- is significantly different, but not in Figure 1D, especially when the difference between the means actually appears greater in Figure 1D than in Figure 1E. Similar trends are observed in Figure 1C, Figure 1F, Figure 2A, Figure 2B…Statistical analysis is impacted by sample size and it raises concerns about whether additional animals in certain groups would change the statistical analysis and thus the central conclusions of this study. Phenotypes in animal models are complex but given that these reported differences are central to the conclusions of the study, this is an important issue that should be addressed by the authors. In lines 178-179, the author’s acknowledge, “This level of variation in hyperpermeability is similar to the variation in inflammation we previously described in CV sox10 mutants.” However, the fact the variation was “previously described” doesn’t satisfy the discrepancy of the observed vs reported statistical differences in the data.

-The authors should report the P-values for all data and the means in the box and whisker plots.

-The authors should repeat experiments with GF WT and GF sox10 -/- in Figure 1C, Figure 1D, Figure 1E, Figure 1F, and others (also see comments below) using additional independent animal groups to demonstrate reproductivity of the reported data. If the data are reproducible, the differences become clearer and the data should remain statistically significant, which will strengthen the conclusions of this study.

-In lines 212-215, the author’s state, “The observation that hyperpermeability in sox10 mutants is independent of microbiota and preceded intestinal neutrophil influx by at least a day strongly suggests that ENS-mediated regulation of permeability is required to prevent microbial dysbiosis and intestinal inflammation.” I’m not fully convinced by the 5 dpf differences in Fig. 2A and Fig. 2B. Specifically, the authors report that Fig. 2A is not significant and Fig. 2B is significant, although they both appear to be trending in the same direction and thus the sentence, “…strongly suggests that ENS-mediated regulation of permeability is required to prevent microbial dysbiosis and intestinal inflammation” is an overstatement. The authors should repeat the experiments in Fig. 2A and Fig. 2B to better support their conclusions.

-In Figure 4, the data suggest that the microbiota can also affect intestinal pH. In Figure 5 and Figure 6, the authors should repeat the experiments with omz and ACTZ using GF zebrafish to determine the contribution of the microbiota on inflammation and overgrowth of Vibrio strains (i.e monocolonized with Vibrio).

-In Fig. 6C, why is the CFU data for CV WT + ACTZ not significantly higher but it is for GFP fluorescence in Fig. 6D? Since GFP fluorescence can be influenced by external factors such as oxygen, which can change due to permeability and inflammation, I would not trust GFP data alone without validation by CFU plating. The authors should address this discrepancy.

- In Fig. 6C and Fig. 6D, the authors compare CV WT to CV WT + OMZ and also compare CV sox10-/- to CV sox10-/- + OMZ and despite an observable drop in CFU in the CV WT + OMZ group, it’s not clear whether the figure is mislabeled or the authors made a mistake in the writing. The authors should perform a statistical analysis to compare the CV WT + OMZ and the CV sox10-/- + OMZ to determine if there is a difference between those two groups.

**Part III – Minor Issues: Editorial and Data Presentation Modifications**

Reviewer #1: 1) The authors do a good job of showing that the ENS contributes to barrier function, transit time, and luminal pH. The experiments in which intestinal pH is corrected in the sox10 background or perturbed (decreased) in the WT background are particularly well-done. Showing that decreased pH is sufficient for pro-inflammatory Vibrio expansion is convincing. However, I believe the authors would be mistaken to not investigate the possible relationship between intestinal pH disregulation and the other sox10 phenotypes described in the manuscript. It seems to me that while this would require additional experiments, it would not require any methodology that is not already present in the study. The questions I would be happy to see answered are as follows:

-If luminal pH in sox10 mutants is corrected to WT levels by omeprazole, does that have an effect on gut transit time or hyperpermeability?

-if luminal pH is decreased in WT animals by ACTZ, does that have an effect on gut transit time or hyperpermeability?

I would like to stress that I don’t see these experiments as absolutely necessary, but I do think they have a potentially high payoff and would only strengthen the paper. Even if the results are negative (luminal pH has no effect on transit time or permeability) they would show that these phenotypes are independent from one another.

2) In figure panels 6H and 6I, the authors point is to show that Vibrio benefits from a lower pH. In my opinion, this is the least convincing point of the paper. My thinking is heavily influenced by the pH readouts used in figure 5. While m-cresol purple appears to give a pH value near 10 for WT intestinal pH, phenol red appears to give a value slightly below 9. The fact that the two dyes do not agree (the CV sox10 values appear to be shifted similarly) suggests that these readouts might only be trustworthy as relative pH values, and not necessarily as absolute values (they are never regarded as absolute values in the manuscript, to be fair). The reason I find this important is that panels 6H and 6I show that Vibrio grows well at pH values as high as 9 and even reasonably well at pH 10. It does not read as a particularly solid mechanism for how Vibrio is able to bloom. Additionally, in vitro conditions are notoriously limited in their ability to provide insight into in vivo phenotypes. How exactly the authors should address this point is unclear to me, but I can offer some rough suggestions. If the authors were to show similar curves for other zebrafish commensals that are present at this larval stage, it could help strengthen the argument, assuming that those commensals are negatively affected by lower pH. A potentially more powerful approach would be to assess the microbial composition of the GI tract in the WT and sox10 backgrounds with and without omeprazole or ACTZ treatment and, if possible, with and without the presence of the pro-inflammatory Vibrio(s). While this approach would not address the mechanism of Vibrio expansion directly, it would serve to show how the treatments affect microbial composition, which may help to elucidate relationships between Vibrio and other commensals under these conditions. Another angle of approach that may be useful is looking at the anti-inflammatory bacteria used in a previous sox10 study (Rolig et al., 2017, PLOS Biology, reference #9 in this manuscript). In that study, E. coli and Shewanella isolates were used to decrease inflammation in CV sox10 mutants. Does addition of these isolates correct the luminal pH or is their anti-inflammatory effect mediated by some other mechanism?

3) In lines 493-495, the authors argue that the impaired intestinal permeability of sox10 mutants could be an underlying mechanism of the luminal pH decrease. Is this not in opposition to the observation that omeprazole corrects the pH back to WT levels? This would suggest that the pH change is an active process mediated by proton pumps, not a passive process mediated by solute leakage from the host.

4) My last point contains a number of related observations about the work. I find it to be an exciting possibility that the ENS regulates the pH of the gastrointestinal tract and I believe that there could be significant implications if this is true in mammals, especially humans. The manuscript does a good job of leading the reader to understand what some potential implications of these findings are, but I think it is lacking in fair comparison of host physiologies. It appears to me that the GI tract of larval zebrafish is quite different from that of a human. Seeing as this study is at least in part funded by the NIH, I would like to see more discussion throughout the text of how this study relates to the human (or even just mammalian) GI tract. For instance, the proton pump ATP4A is responsible for gastric acidity in humans and is not expressed in the GI tract outside of the stomach, as far as I am aware. How does this compare the localization of proton pumps in larval zebrafish? The vast majority of microbes that inhabit the human host reside specifically within the large intestine, where they can reach loads as high as 1011-13 CFUs/g contents. In the larvae studied here, it appears that the highest microbial burden is carried in the bulb. What are the implications of that? In figure 3, tight junctions are found to impaired specifically in the colon-like region but the implication of that is not discussed. There are more specific instances of physiological differences that could be mentioned but my intention is not for the manuscript to be a defense of zebrafish as a model. What I would like to see is a greater effort to address the relatively large differences between the larval zebrafish GI tract and the mammalian GI tract, with special considerations made for spatial (stomach, small intestine, large intestine, etc.) and environmental differences.

Finally, I would like to thank the authors for this work. It was a clean and enjoyable read.

Reviewer #2: 1. What is EM (line 365)?

2. While occludin binding to ZO1 was decreased in GF sox10 compared to CV WT, it is unclear (statistics not showed) whether there was a statistically significant difference between CV WT and GF WT, or between GF WT and GF sox10 (Fig. 3A). Therefore, it is unclear whether the hyperpermeability phenotype is independent of the microbiota status, at least in the ileum-like region.

3. To substantiate the claim that “reduced luminal pH of sox10 mutants induces the hyperinflammatory state”, the authors should add the statistics of the comparison of WT and sox10+omz, which will show whether reducing luminal pH is sufficient to reverse the phenotype.

Reviewer #3: -In lines 64-65, the authors say, “Using zebrafish lacking an ENS due to a homozygous null mutation in the sox10 gene we discovered that the ENS constrains intestinal microbiota composition and intestinal inflammation, as measured by intestinal neutrophil infiltration.” What is sox10? What it’s function? The authors should add a sentence or two to better introduce sox10. This is important given that sox10 is central to the study.

-In lines 182-183, the authors write, “We observed that even in the absence of microbiota, sox10 mutants had significant levels of fluorescence in the aorta compared to GF WTs (Fig 1E and F).” What does “significant levels” mean? Significant higher or lower levels? The authors call out Fig. 1E and Fig. 1F but in Fig. 1F the GF data is not significant. The authors needs to explain this discrepancy and make the writing more clear.

-In lines 174-176, the writing is confusing. The authors state, “CV WTs did not have detectable fluorescence in the vasculature. In contrast, CV sox10 mutants had measurable levels of fluorescence in the aorta compared to CV WTs (Fig 1E and F).” When the authors state, “CV WTs did not have detectable fluorescence in the vasculature.” What figure is that referring to? If they are referring to Fig 1E and F, then what are the data points shown in these figures for CV WT? Presumably, if there are data points for CV WT, then the fluorescence was measurable, correct? The authors need to be more clear about exactly which Figures they are calling out in the writing.

-In Figure 4B and D, the authors should also perform as statistical analysis to compare the CV WT and GF WT and CV sox10-/- group to the GF sox10-/- group to determine the contribution of the microbiota in these groups.

-In lines 315-316, the author’s write “We hypothesized that the hyperinflammation seen in sox10 mutant intestines could be corrected by increasing the luminal pH to levels similar to WTs.” This sentence, needs more context Why did you hypothesize that?

-In lines 316-317, the authors wrote, “We first verified that zebrafish intestinal cells have proton pumps using single cell RNA-seq data from intestinal tissue [58].” This sentence is confusing because it reads like the authors did an experiment for this study (which is not shown) but they are citing another study. It seems more appropriate for the sentence to read, “We have found that zebrafish intestinal cells have proton pumps using single cell RNA-seq data from intestinal tissue [58].

-In line 363-365, the authors state “Omeprazole reduced Vibrio-GFP abundance in both CV WTs and sox10 mutants as shown by GFP-CFU (Fig 6C) and fluorescence (Fig 6D)” and in lines 367-368, “DMA treatment decreased intestinal abundance of Vibrio-GFP (S2 Figs C and D)” However, the data shown in Figure 6C,D and Figure S2C indicates that there is no statistical difference in Vibrio CFU/intestine between CV WT and CV WT + OMZ and CV WT and CV WT + DMA, respectively. It seems unlikely they would have mislabeled both of the figures. There appears to be a large difference in CFU (close to one log) between these groups. The authors should explain this discrepancy.

-Since DMA has antibacterial activity in vitro independent of zebrafish, it’s not clear why this data is included in this study since it’s not clear whether the in vivo data are due to acidity other to direct antibacterial activity of DMA.

PLOS authors have the option to publish the peer review history of their article (what does this mean?). If published, this will include your full peer review and any attached files.

Reviewer #1: No

Reviewer #2: No

Reviewer #3: No
---

## [Editor Report · Decision Letter 1]

7 Jan 2022

Dear Dr. Eisen,

We are pleased to inform you that your manuscript 'Enteric nervous system modulation of luminal pH modifies the microbial environment to promote intestinal health' has been provisionally accepted for publication in PLOS Pathogens.

Best regards,

Andreas J Baumler

Associate Editor

PLOS Pathogens

Nina Salama

Section Editor

PLOS Pathogens

Kasturi Haldar

Editor-in-Chief

PLOS Pathogens

orcid.org/0000-0001-5065-158X

Michael Malim

Editor-in-Chief

PLOS Pathogens

orcid.org/0000-0002-7699-2064
---

## [Editor Report · Acceptance letter]

18 Jan 2022

Dear Dr. Eisen,

We are delighted to inform you that your manuscript, "Enteric nervous system modulation of luminal pH modifies the microbial environment to promote intestinal health," has been formally accepted for publication in PLOS Pathogens.

Best regards,

Kasturi Haldar

Editor-in-Chief

PLOS Pathogens

orcid.org/0000-0001-5065-158X

Michael Malim

Editor-in-Chief

PLOS Pathogens

orcid.org/0000-0002-7699-2064